# N$_2$O Emission from Partial Nitrification and Full Nitrification in Domestic Wastewater Treatment Process

Pengzhang Li [1], Yongzhen Peng [2,*], Shuying Wang [2] and Yue Liu [2]

[1] Department of Municipal Engineering, Yancheng Institute of Technology, Yancheng 224051, China
[2] Engineering Laboratory for Advanced Municipal Wastewater Treatment and Reuse Technology, Engineering Research Center of Beijing, Beijing University of Technology, Beijing 100124, China
* Correspondence: pyz@bjut.edu.cn; Tel./Fax: +86-10-67392627

**Abstract:** Using actual domestic wastewater as the research object, nitrogen compounds and their combinations were added to different nitrification (partial nitrification, full nitrification) processes to investigate nitrous oxide (N$_2$O) emission and its nitrification mechanisms. The presence of influent NH$_4^+$ was the driving force of N$_2$O emission during nitrification. Compared with full nitrification, NO$_2^-$ in partial nitrification more readily generated N$_2$O by denitrification. Under the proportional gradient of NH$_4^+$-N:NO$_2^-$-N/NO$_3^-$-N, 30:0, 20:10, 10:20, and 0:30, total N$_2$O emissions during partial nitrification were 2.81, 11.30, 65.20, and 11.67 times greater than the total N$_2$O emissions during full nitrification. Full nitrification was more beneficial to N$_2$O emission reduction. This provides a control strategy for N$_2$O emission reduction in wastewater treatment processes under the background of reducing the production of greenhouse gases.

**Keywords:** N$_2$O; domestic wastewater; partial nitrification; full nitrification; greenhouse gas reduction





## 1. Introduction

Nitrous oxide (N$_2$O) is a potent greenhouse gas whose greenhouse effect exceeds CO$_2$ by ~300 times [1]. N$_2$O is also a potential ozone depleting substance (ODS) [2]. Wastewater biological treatment is an anthropogenic source of N$_2$O emissions, and both nitrification and denitrification processes produce N$_2$O [3–6]. Full nitrification involves two steps: NH$_4^+$ oxidization to NO$_2^-$ via NH$_2$OH utilizing Ammonia Oxidizing Bacteria (AOB) followed by subsequent oxidation of NH$_2$OH to NO$_2^-$ by AOB (the energy generation step); Secondly, NO$_2^-$ is further oxidized to NO$_3^-$ using Nitrite Oxidizing Bacteria (NOB). In this process, ammonia mono-plus oxidase (AMO) catalyzes the oxidation of NH$_3$ to NH$_2$OH and O$_2$ acts as the electron acceptor [7]. Hydroxylamine oxidase (HAO) catalyzes the oxidation of NH$_2$OH to NO$_2^-$ and O$_2$ acts as the main electron acceptor [8]. In the nitrification process, there are two possible pathways for N$_2$O emission as a byproduct: (1) N$_2$O is produced during autotrophic denitrification by AOB, NO$_2^-$ acts as an electron acceptor, and is converted into N$_2$O via NO using nitrite reductase and nitric oxide reductase [9,10]; (2) N$_2$O is produced by the incomplete oxidation of NH$_2$OH [11–13].

A few studies have reported some operational control factors related to N$_2$O emission regarding full nitrification, such as dissolved oxygen (DO), temperature, pH, Sludge Retention Time (SRT), salinity, and toxic substances [14–21]. However, fewer studies on N$_2$O emission during partial nitrification have been published. Most studies were conducted using culture medium and artificial wastewater, which does not accurately simulate the complex actual nitrification.

In this study, a Sequencing Batch Reactor (SBR) and specific test rules explored N$_2$O emission and some nitrification mechanisms using partial and full nitrification of sludge cultivated with actual domestic wastewater. These results provide a theoretical basis for the control of N$_2$O emission in wastewater treatment.

## 2. Methods

### 2.1. Sludge and Wastewater

Two 12L SBRs were used for cultivation of partial and full nitrification sludge and the index of domestic wastewater inflow is shown in Table 1. The traditional full nitrification operation mode was as follows. The average operational cycle was 420 min, which included feeding (30 min), aeration (240 min), anoxic denitrification (120 min), and settling (30 min), 3 cycles per day, with the DO was 2 mg/L and Mixed Liquid Suspended Solids (MLSS) remaining at about 3000 mg/L. SRT was 20 days, and the rate of nitrate accumulation during full nitrification was about 99%. The partial nitrification operation mode was similar to that for full nitrification, except the DO was 1 mg/L and running temperature was 30 °C, SRT was 11 days; the rate of nitrite accumulation during partial nitrification was about 98%. After anoxic denitrification, the effluent's composition, consisting of $NH_4^+$-N, $NO_2^-$-N, and $NO_3^-$-N were all below 1 mg/L. The full nitrification and partial nitrification sludge taken from the SBRs was aerated for 12 h, then washed 3 times with deionized water repeatedly, for batch testing. The effluent's COD was less than 50 mg/L and could not be oxidized for longer.

**Table 1.** The quality of real domestic wastewater.

|  | COD (mg/L) | $NH_4^+$-N (mg/L) | $NO_2^-$-N (mg/L) | $NO_3^-$-N (mg/L) | TN (mg/L) | pH | Alkalinity |
|---|---|---|---|---|---|---|---|
| minimum | 88 | 39.6 | 0 | 0 | 56.4 | 6.9 | 262 |
| maximum | 276 | 91.2 | 2.8 | 1.2 | 98.5 | 7.7 | 343 |
| average | 182 | 65.4 | 1.4 | 0.6 | 77.4 | 7.3 | 303 |

### 2.2. Batch Test Rules

The experimental batch test reactor is shown in Figure 1. The effective volume of the reactor was 3 L. At the beginning of each batch test, 1 L of concentrated full nitrification and partial nitrification sludge was added into the reactor, followed by 2 L of wastewater, and the MLSS were controlled at 3000 mg/L. Nitrogen compounds, DO and pH levels were then adjusted for the batch set upon commencing operation (Table 2). The running time for batch tests was 180 min.

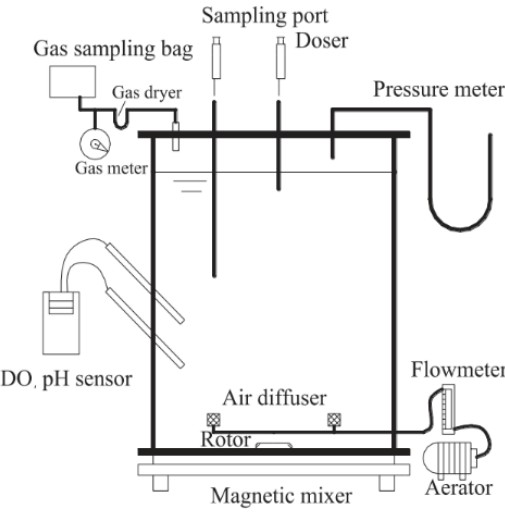

**Figure 1.** $N_2O$ emission during partial nitrification (partial nitrification effluent +$NH_4Cl$-N/$NaNO_2$-N, DO = 0.5 mg/L).

**Table 2.** Batch test rules.

| Batch Test Number | Sludge—Water Mixture Type | Initial pH | Reaction Time (h) | $NH_4^+$-N (mg/L) | $NO_2^-$-N (mg/L) | $NO_3^-$-N (mg/L) | DO (mg/L) |
|---|---|---|---|---|---|---|---|
| 1 | partial nitrification sludge + effluent | 7.5 | 3 | 30 | 0 | | 0.5 |
| | partial nitrification sludge + effluent | 7.5 | 3 | 20 | 10 | | 0.5 |
| | partial nitrification sludge + effluent | 7.5 | 3 | 10 | 20 | | 0.5 |
| | partial nitrification sludge + effluent | 7.5 | 3 | 0 | 30 | | 0.5 |
| 2 | full nitrification sludge + effluent | 7.5 | 3 | 30 | | 0 | 0.5 |
| | full nitrification sludge + effluent | 7.5 | 3 | 20 | | 10 | 0.5 |
| | full nitrification sludge + effluent | 7.5 | 3 | 10 | | 20 | 0.5 |
| | full nitrification sludge + effluent | 7.5 | 3 | 0 | | 30 | 0.5 |
| 3 | full nitrification sludge + effluent | 7.5 | 3 | | 20 | | 0.5 |
| | full nitrification sludge + effluent | 7.5 | 3 | | 20 | | 1 |
| | full nitrification sludge + effluent | 7.5 | 3 | | 20 | | 2 |
| | full nitrification sludge + effluent | 7.5 | 3 | | 20 | | 3 |

### 2.3. Detection Method

COD, $NH_4^+$-N, $NO_2^-$-N, $NO_3^-$-N were measured according to methods previously described [22]. DO, pH, T were measured by an oxygen, pH and temperature meter (WTW 340i, WTW Company, Munich, Germany). The Mixed Liquid Suspended Solids concentration was measured at the beginning and at the end of each test to obtain an average value, which was used for the calculation of the $NH_4^+$-N oxidation rate, $NO_x^-$-N production rate and $N_2O$ emission rate. The total $N_2O$ production consists of the $N_2O$ emitted in the gaseous phase (emission-gas $N_2O$) and the $N_2O$ dissolved in the mixed liquid phase (dissolved $N_2O$). The $N_2O$ concentrations in gas samples were analyzed in triplicate using a gas chromatograph (Agilent 6890N, Santa Clara, CA, USA). The overhead space method was used to analyze the dissolved $N_2O$. Water and $N_2O$ samples were taken at 30-min intervals.

### 3. Results and Discussion

#### 3.1. $N_2O$ Emission in Partial Nitrification Process

Figure 2 shows the variations of $N_2O$ emissions under different $NH_4^+$-N and $NO_2^-$-N ratios during partial nitrification (partial nitrification's effluent +NH4Cl/NaNO2). As shown in Figure 2a–c, the maximum $N_2O$ emission occurred when $NH_4^+$-N was about to be oxidized completely, and its maximum value was 1.20 mg/L, 1.37 mg/L, and 1.48 mg/L. When the ratios of $NH_4^+$-N to $NO_2^-$-N were 30:0, 20:10, and 10:20, the time for $N_2O$ to reach its highest level decreased, and the $N_2O$ emission rates increased; they were 0.16, 0.30, and 0.49, respectively (in mgN/(gMLSS·L·h)). This indicated that the initial emission rate of $N_2O$ increased with added $NO_2^-$ in the presence of $NH_4^+$-N. This was due to the electrons provided by the $NH_4^+$ oxidation process being used for autotrophic denitrification of AOB in the partial nitrification process.

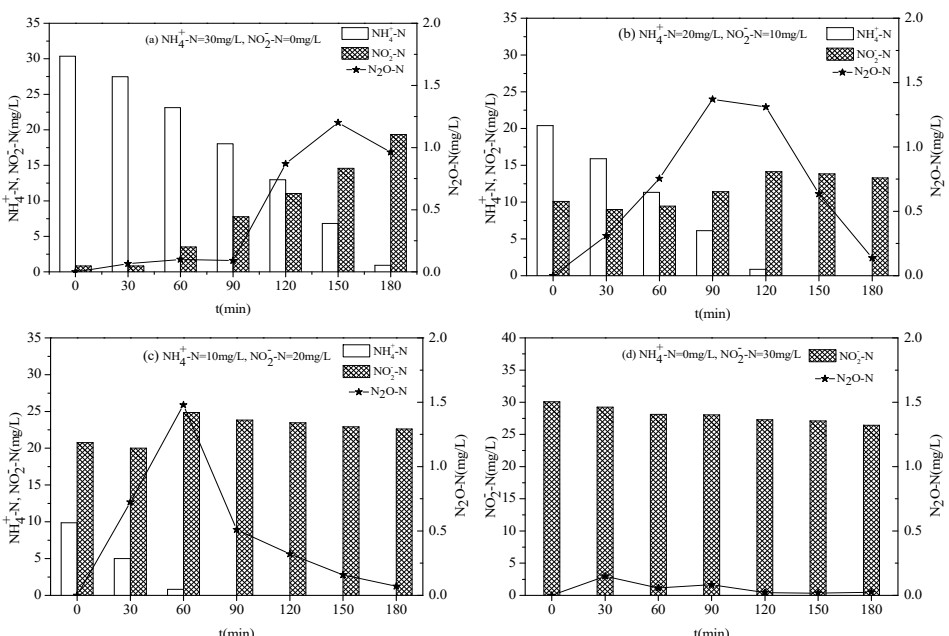

**Figure 2.** N$_2$O emission during partial nitrification (partial nitrification effluent +NH$_4$Cl-N/NaNO$_2$-N, DO = 0.5 mg/L).

When NH$_4^+$ oxidation provided electrons, higher concentrations of electron acceptor NO$_2^-$ led to increased autotrophic denitrification of AOB. As shown in Figure 2b,c, NO$_2^-$ concentrations decreased with NH$_4^+$ oxidation in the first 30 min of the reaction. However, in the absence of NH$_4^+$, as shown in Figure 2d, even a NO$_2^-$ concentration of 30 mgN/L did not result in N$_2$O emission above 0.15 mg/L. This reaction resulted in denitrification of only 3.5 mgN/L of NO$_2^-$. The lack of BOD (effluent) suggested its electron source might be internal organic matter (PHB) stored in AOB sludge [23,24], hydrogen, and pyruvate [25]. In addition, under four different ratios of NH$_4^+$-N and NO$_2^-$-N, the total production of N$_2$O was 3.29, 4.52, 3.26, and 0.35 mgN/L, respectively. N$_2$O production was maximized when NH$_4^+$-N and NO$_2^-$-N ratios were 20:10. This was due to the presence of electron acceptor NO$_2^-$ (10 mgN/L), and contrasted with the minimal levels observed with the ratio of 30:0 (Figure 2b). Compared with 10:20, the reaction in Figure 2b had more electron donors from NH$_4^+$.

### 3.2. N$_2$O Emission in Full Nitrification Process

Figure 3 shows the variation of N$_2$O emission during full nitrification (full nitrification effluent +NH$_4$Cl/NaNO$_2$) under different NH$_4^+$-N and NO$_3^-$-N ratios. As shown in Figure 3a–c, with decreasing NH$_4^+$-N and increasing NO$_3^-$-N, the time it took for N$_2$O to maximize decreased; however, the N$_2$O emission rates also decreased (0.051, 0.029, and 0.005, mgN/(gMLSS·L·h)). When the ratios of NH$_4^+$-N to NO$_3^-$-N were 30:0, 20:10, and 10:20, the maximum yields of N$_2$O were 0.46, 0.18, and 0.02 mgN/L, respectively. As shown in Figure 3a–c, the maximum production of N$_2$O occurred as NH$_4^+$-N was oxidized, which was the same as for the N$_2$O emissions in the AOB enrichment system. Figure 3 also shows that under the four proportional gradients, the total N$_2$O production was 1.17, 0.40, 0.05, and 0.03 (mgN/L), which indicated that with full nitrification (where AOB and NOB co-exist), the production of N$_2$O mainly depended on the initial concentration of NH$_4^+$, rather than the concentration of NO$_3^-$. In addition, as shown in Figure 3d, in the absence of COD, a very small amount of N$_2$O was still generated when NO$_3^-$ was added to the system, which may be caused by the denitrification of using internal organic matter in sludge.

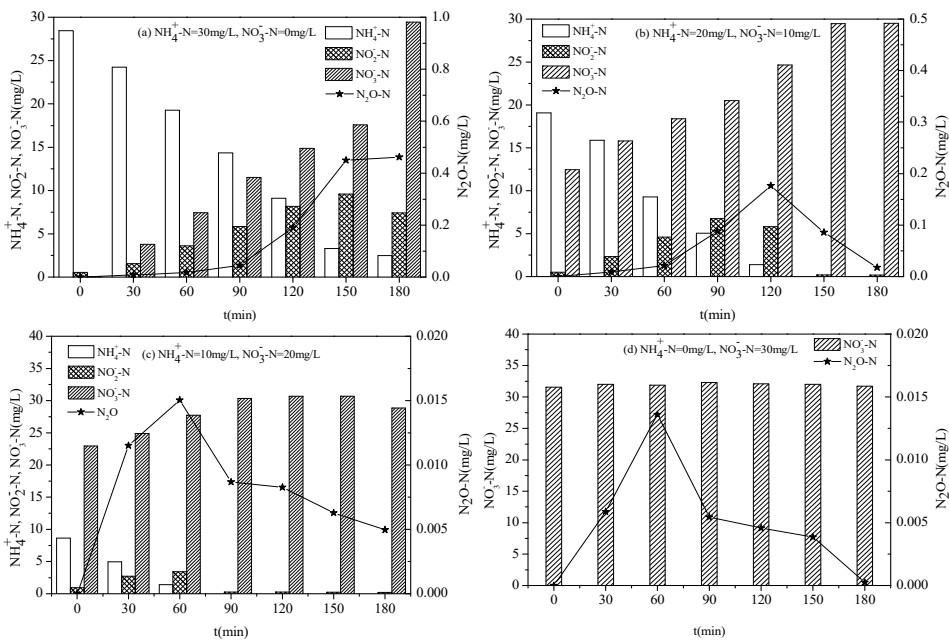

**Figure 3.** N$_2$O emission during full nitrification (full nitrification effluent +NH$_4$Cl-N/NaNO$_2$-N, DO = 0.5 mg/L).

Figure 4 shows the variation of N$_2$O emissions during full nitrification (full nitrification effluent +NaNO$_2$ 20 mgN/L). This batch test added 20 mgN/L NaNO$_2$ to the nitrification system effluent and investigated the oxidation of NO$_2^-$ by NOB to produce N$_2$O under different DO concentrations. As shown in this figure, N$_2$O reached its maximum value after 30 min, and its maximum production decreased with increasing DO, with values of 0.054 mg/L, 0.051 mg/L, 0.014 mg/L, and 0.007 mg/L. The NO$_2^-$ oxidized to NO$_3^-$ within 60 min. After 60 min, no NO$_2^-$ remained in the system (data not shown), but the production of N$_2$O was not 0, which indicated that during full nitrification, there was still N$_2$O production during NO$_2^-$ oxidation to NO$_3^-$ by NOB. Although the amount was very small, this N$_2$O may come from microorganisms using endogenous substances to provide electrons for denitrification. During the reaction, the total production of N$_2$O was 0.14 mg/L, 0.09 mg/L, 0.04 mg/L, and 0.03 mg/L for the four different DO gradients, and the proportion of N$_2$O in influent NO$_2^-$-N was 0.70%, 0.45%, 0.20%, and 0.15%. The percentage was smaller than N$_2$O production in an AOB enriched system (Figure 2d), which was 1.17% (when DO = 0.5 mg/L) and indicated that NO$_2^-$ in an AOB enriched system was more readily denitrified than with full nitrification.

As shown in Figures 2 and 3, under the same ratio of NH$_4^+$-N to NO$_2^-$-N and NO$_3^-$-N (DO = 0.5 mg/L), and under four proportional gradients, in the AOB system with NO$_2^-$, the total N$_2$O emissions were 2.81, 11.30, 65.20, and 11.67 times greater than the total N$_2$O emission during full nitrification, which indicated that under the same conditions of influent NH$_4^+$-N, partial nitrification produced more N$_2$O than full nitrification. During partial nitrification, NO$_2^-$ was more involved in autotrophic denitrification of AOB as a product, while during full nitrification, NO$_3^-$ would not participate in autotrophic denitrification as a nitrification product. Moreover, NO$_2^-$ further oxidized to NO$_3^-$ by NOB as an intermediate product of full nitrification; therefore, only a small amount of NO$_2^-$ was involved in autotrophic denitrification by AOB.

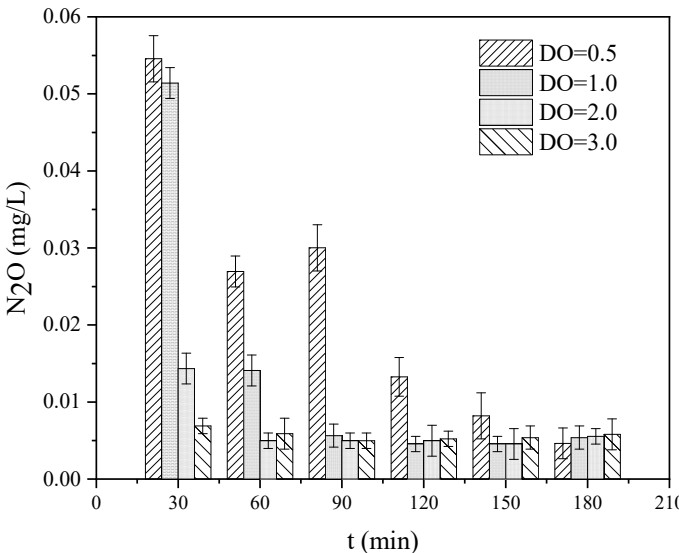

**Figure 4.** $N_2O$ emission during full nitrification (full nitrification effluent +NaNO$_2$-N 20 mgN/L).

## 4. Conclusions

The $N_2O$ emission in partial and full nitrification of actual domestic wastewater was investigated under specific conditions and yielded the following conclusions:

(1) The presence of influent $NH_4^+$ was the driving force of $N_2O$ emissions in full and partial nitrification processes.

(2) Compared with full nitrification, $NO_2^-$ was more likely to participate in denitrification during partial nitrification to produce $N_2O$.

(3) Under four proportional gradients, the total production of $N_2O$ during partial nitrification was 2.81, 11.30, 65.20, and 11.67 times greater than the total $N_2O$ production during full nitrification.

**Author Contributions:** P.L.: Investigation, Sampling, Data curation, Writing – original draft; Y.L.: review & editing; P.L.: Investigation, Sampling, Data curation, Writing – original draft; Y.L.: review & editing; S.W.: Resources, Funding acquisition; Y.P.: Supervision, Visualization. S.W.: Resources, Funding acquisition; Y.P.: Supervision, Visualization. All authors have read and agreed to the published version of the manuscript.

**Funding:** This research project was financially supported by National Key R&D Program of China (Grant No. 2021YFC3200601) and Yancheng Institute of Technology's Start-up Research Fund.

**Data Availability Statement:** The original data is backed up on my computer, ready for investigation.

**Conflicts of Interest:** The authors declare that they have no known competing financial interests or personal relationships that could have appeared to influence the work reported in this paper.

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
