# Peer review of "N2O Emission from Partial Nitrification and Full Nitrification in Domestic Wastewater Treatment Process"

_water, doi:10.3390/w14203195_

Round 1

Reviewer 1 Report

This paper reports the observation of N2O emission from partial nitrification and full nitrification in domestic wastewater treatment process,under the background of carbon neutrality, it has certain academic and practical significance. Below are some of mine comments, might help the paper more precise.

1. This paper does not consider the portion of organic-N in the wastewater, which easily converts to NH3-N and increases the amount of NH3-N in the wastewater when hydrolysis happens. When determining the ammonia oxidation rate, it would better use inorganic NH3-N as feed, instead of using raw wastewater. Besides, NH3-N is also utilized by microorganisms for cell growth. There must be some loss of NH3-N during the treatment. The conversion of NH3-N is complicated, but the paper is too much simplified the other conditions that should be considered.

2. When measuring NO2- production, the loss of nitrogen caused by heterotrophic denitrification should be considered.

3. DO level was controlled by aeration. At a higher DO level, a higher aeration rate is required and causes more N2O stripped out from the liquid. However, the higher flow rate of the air also dilutes the concentration of N2O. In this paper, the effect of aeration to N2O emission was not included. The measured N2O might not be correct.

4. Ammonia oxidation rate is highly sensitive to temperature. The temperature of the experiment was fixed at 30℃, which was higher than the typical wastewater treatment temperature. This paper deals with real wastewater, it is better to choose a normal treatment temperature for the experiment. What was the SRT of the treatment system?

5. The overhead space method was used to analyze the dissolved N2O is not well described.

Author Response

1.This paper does not consider the portion of organic-N in the wastewater, which easily converts to NH3-N and increases the amount of NH3-N in the wastewater when hydrolysis happens. When determining the ammonia oxidation rate, it would better use inorganic NH3-N as feed, instead of using raw wastewater. Besides, NH3-N is also utilized by microorganisms for cell growth. There must be some loss of NH3-N during the treatment. The conversion of NH3-N is complicated, but the paper is too much simplified the other conditions that should be considered.

Answer: Thank you very much for your question. There is organic nitrogen in real domestic wastewater absolutely. However, in our experiment, we measured that 50 mg COD contained 1 mg organic nitrogen in real domestic wastewater. The domestic wastewater which we used in the experiment contains about 200 mg/L COD, so the organic nitrogen which could be converted to NH3 was about 4 mg/L. Similarly, NH3, as a growth source of microorganisms, was about 2~3 mg/L. As you mentioned, these two cases would have effect on NH4+-N oxidation ratio, but on the other hand, the converted NH3 and the used NH3 can offset each other in a sense. So in the presence of COD, organic nitrogen may have no effect on NH4+-N oxidation ratio. Besides, it might be more practical significance to study N2O by using real domestic wastewater.

2.When measuring NO2- production, the loss of nitrogen caused by heterotrophic denitrification should be considered.

Answer: We recognize that microbial heterotrophic denitrification can actually cause nitrogen in the form of gases (N2, NO, N2O) during nitrifying process, especially under the condition of low DO, there will be more nitrogen losses caused by heterotrophic denitrification. However, N2O produced by heterotrophic denitrification was also recorded in the test of N2O, and nitrogen losses caused by heterotrophic denitrification may be very few during nitrifying process.

3.DO level was controlled by aeration. At a higher DO level, a higher aeration rate is required and causes more N2O stripped out from the liquid. However, the higher flow rate of the air also dilutes the concentration of N2O. In this paper, the effect of aeration to N2O emission was not included. The measured N2O might not be correct.

Answer: Thank for your question. Indeed, we are aware that more N2O will be stripped from water by increasing aeration rate, then cause N2O concentration (Cd(t)) in the liquid to reduce. These stripped N2O can be completely collected in air bag, increasing aeration rate can also increase the mixed gases volume (V), from equation (3) we know re (v/v) will reduce by increasing V, but N2O concentration depends on re and collected gases volume (V) in air bag. In addition, we use gas chromatography to measure N2O in our experiment, it is a precise method.

4.Ammonia oxidation rate is highly sensitive to temperature. The temperature of the experiment was fixed at 30℃, which was higher than the typical wastewater treatment temperature. This paper deals with real wastewater, it is better to choose a normal treatment temperature for the experiment. What was the SRT of the treatment system?

Answer: We agree with reviewer’s comments about ammonia oxidation rate being highly affected by temperature. This experiment was done at real-time room temperature of 30C, because most Chinese wastewater treatment plants’ water temperature ranges from 28C~32C. We are not purposely control temperature in the tests. Moreover, it also helps to simulate WWTPs’ N2O production during nitrifying process in summer.

1m3 of mixed liquor was wasted per cycle, giving rise to a sludge retention time (SRT) of 20 days, and the hydraulic retention time (HRT) was controlled at 15h. Now the information is included in the revised text.

5.The overhead space method was used to analyze the dissolved N2O is not well described.

Answer: We did not describe head space method clearly due to the word limitation of the text. N2O dissolved in the activated sludge was detected using head space method: Take 5 mL mixture using 20 mL syringe, then add 0.5 mL solution of HgCl2 (1000 mg/L) to the syringe to restrain the activated sludge and extract 5 mL N2 from N2 bag using syringe, put the syringe into thermostatic oscillator to shake at 30°C for 0.5 hour. Finally, use gas chromatograph to detect N2O gas concentration (ppm) in the head space of this syringe, then calculate dissolved N2O quality concentration (mg/L) according to Henry’s law.

Reviewer 2 Report

The manuscript describes a study investigating the production of nitrous oxide during nitrogen conversion/removal in wastewater treatment systems. The issue of nitrous oxide production during biological wastewater treatment is relevant, but the manuscript is not clear on what is particularly novel and does not fully explain the methods used. Additional detail is needed before this manuscript is ready for acceptance.

Specific comments include:

Line 34. I believe the term “denitrification of AOB” should be “denitrification by AOB”.

Line 46. The last line of the introduction is out of place. The introduction has no discussion of carbon as it relates to biological nitrogen transformations. The introduction needs to include discussion of carbon if this comment is to remain.

Line 57. The methods do not clearly explain the seed reactors used in the study. In particular the conditions of the partial nitrification system are not understood. The statement that effluent contained all nitrogen species below 1 mg/L implies that the seed reactors were both fully nitrifying. More explanation is needed to describe the two seed reactors and how the biomass in each differed.

Line 58. The description of sludge washing is unclear. What was this for? How was it done? More explanation is required.

Line 59. No method for COD appears to be provided. A method should be described.

Line 59. What is the basis for saying that the effluent COD can no longer be oxidized?

Table 2. What is “mud water mixture type”?

Table 2. The word “effluent” in all of the names in the “mud water mixture type” column does not make sense. This should be referring to the type of biomass used for the test, not the effluent from the reactors.

Line 91. The wording in the results and discussion is misleading. The manuscript describes results as “during” partial or full nitrification. This does not appear accurate for the “partial nitrification” results as the experimental results actually show tests that completely nitrify. It appears the intention is to show difference in N20 generation between sludges acclimated under different conditions, but during the same test conditions. This should be clarified.

Line 94. The unit “mg/L” used to explain total N20 emissions (both gaseous and dissolved) is difficult to understand. What does the liter component refer to? Rector volume? Volume of feed? Volume of off-gas? This unit needs to be more specific. Also, it would be helpful to present the N20 production as mass proportion of the influent nitrogen rather than volumetric flow. This way the rate of N20 production can better be understood based on the mass balance of nitrogen in the system.

Figure 1. Use the same scale for each vertical axis on all plots in the same figure.

Line 164. The conclusions section is too brief. Based on mention of carbon and COD in the introduction and methods, there should be some discussion of these if carbon is important to the work.

Line 173. Disagree with the last sentence. Full nitrification does not lead to a reduction in N20 production. The results suggest that when using a sludge acclimated to “partial nitrification” conditions more N20 is generated.

Author Response

The manuscript describes a study investigating the production of nitrous oxide during nitrogen conversion/removal in wastewater treatment systems. The issue of nitrous oxide production during biological wastewater treatment is relevant, but the manuscript is not clear on what is particularly novel and does not fully explain the methods used. Additional detail is needed before this manuscript is ready for acceptance.

Specific comments include:

Line 34. I believe the term “denitrification of AOB” should be “denitrification by AOB”.

Answer: We agree with your rigorous suggestions, "denitrification of AOB" is changed to "denitrification by AOB".

Line 46. The last line of the introduction is out of place. The introduction has no discussion of carbon as it relates to biological nitrogen transformations. The introduction needs to include discussion of carbon if this comment is to remain.

Answer: This sentence is changed to "These results provide a theoretical basis for the control of N2O emission in wastewater treatment ".

Line 57. The methods do not clearly explain the seed reactors used in the study. In particular the conditions of the partial nitrification system are not understood. The statement that effluent contained all nitrogen species below 1 mg/L implies that the seed reactors were both fully nitrifying. More explanation is needed to describe the two seed reactors and how the biomass in each differed.

 Answer: Thank for your rigorous suggestions, this paragraph is changed to "…SRT was 20 days, the rate of nitrate accumulation during full nitrification is about 99%. The partial nitrification operation mode works the same way as full nitrification, except the DO at 1 mg/L and running temperature was 30℃, SRT was 11 days, the rate of nitrite accumulation during partial nitrification is about 98%. After anoxic denitrification, the effluent’s composition, …".

Line 58. The description of sludge washing is unclear. What was this for? How was it done? More explanation is required.

Answer: We agree with your rigorous suggestions, this paragraph is changed to "…. The full nitrification and partial nitrification sludge taken from those SBRs was aerated for 12h, then washed using deionized water by repeatedly, they will be applied to batch tests…. " .

Line 59. No method for COD appears to be provided. A method should be described.

Answer: "COD, NH4+-N, NO2--N, NO3--N were measured according to methods previously described [22]. "

Line 59. What is the basis for saying that the effluent COD can no longer be oxidized?

Answer: After partial nitrification and full nitrification, then continue to aeration for a long time, the COD cannot be reduced, and we can infer it is inert organic matter.

Table 2. What is “mud water mixture type”?

Answer: This sentence is changed to "sludge -water mixture type".

Table 2. The word “effluent” in all of the names in the “mud water mixture type” column does not make sense. This should be referring to the type of biomass used for the test, not the effluent from the reactors.

Answer: Thank for your rigorous suggestions, in the whole batch tests, we used a mixture of partial nitrification and full nitrification sludge and effluent to test, please see "2.2. Batch test rules".

Line 91. The wording in the results and discussion is misleading. The manuscript describes results as “during” partial or full nitrification. This does not appear accurate for the “partial nitrification” results as the experimental results actually show tests that completely nitrify. It appears the intention is to show difference in N2O generation between sludges acclimated under different conditions, but during the same test conditions. This should be clarified.

Answer: In batch tests, The characteristics of N2O emission were investigated respectively from partial and full nitrification process to explored the related mechanisms, and found that:(1) The presence of influent NH4+ was the driving force of N2O emission in full and partial nitrification processes;(2) NO2- was more likely to participate in denitrification during partial nitrification to produce N2O;(3) During full nitrification, NO2- further oxidized to NO3- by NOB as an intermediate product of full nitrification; only a small amount of NO2- was involved in autotrophic denitrification by AOB. Yields were also compared.

Line 94. The unit “mg/L” used to explain total N2O emissions (both gaseous and dissolved) is difficult to understand. What does the liter component refer to? Rector volume? Volume of feed? Volume of off-gas? This unit needs to be more specific. Also, it would be helpful to present the N2O production as mass proportion of the influent nitrogen rather than volumetric flow. This way the rate of N2O production can better be understood based on the mass balance of nitrogen in the system.

Answer: Thank for your rigorous suggestions. The N2O concentrations in gas samples were analyzed in triplicate using a gas chromatograph (Agilent 6890N, Santa Clara, California, USA). The overhead space method was used to analyze the dissolved N2O. we did not describe head space method clearly due to the word limitation of the text. N2O dissolved in the activated sludge was detected using head space method: Take 5 mL mixture using 20 mL syringe, then add 0.5 mL solution of HgCl2 (1000 mg/L) to the syringe to restrain the activated sludge and extract 5 mL N2 from N2 bag using syringe, put the syringe into thermostatic oscillator to shake at 30°C for 0.5 hour. Finally, use gas chromatograph to detect N2O gas concentration (ppm) in the head space of this syringe, then calculate dissolved N2O quality concentration (mg/L) according to Henry’s law. This is to be consistent with the other N's quality (N mg/L), so that we can draw and compare them expediently.

Figure 1. Use the same scale for each vertical axis on all plots in the same figure.

Answer: Thank for your rigorous suggestions, in order to make the picture more appropriate and beautiful, we used the different scale for each vertical axis on all plots in the same figure. We will make further improvements.

Line 164. The conclusions section is too brief. Based on mention of carbon and COD in the introduction and methods, there should be some discussion of these if carbon is important to the work.

Answer: Thank for your rigorous suggestions, we focus on N transformations in this study, COD (in water) is not related to the process of autotrophic nitrification.

Line 173. Disagree with the last sentence. Full nitrification does not lead to a reduction in N2O production. The results suggest that when using a sludge acclimated to “partial nitrification” conditions more N2O is generated.

Answer: Thank for your rigorous suggestions, our conclusion is based on the data presented in this manuscript and our rigorous batch tests.

Round 2

Reviewer 2 Report

The manuscript is improved and can be accepted assuming the following minor comment is addressed.

Line 60. The following sentence is incomplete and does not make sense. Please correct for completeness.
"The full nitrification and partial nitrification sludge taken from those SBRs was aerated for 12h, then washed using deionized water by repeatedly, they will be applied to batch tests."

Author Response

Line 60. The following sentence is incomplete and does not make sense. Please correct for completeness. "The full nitrification and partial nitrification sludge taken from those SBRs was aerated for 12h, then washed using deionized water by repeatedly, they will be applied to batch tests."

Answer: Thank for your rigorous suggestions, this sentence explains how we treat sludge,and  make it free of COD and nitrogen compounds from sludge culture reactors,they will be applied to batch tests for next step . This sentence is changed to "The full nitrification and partial nitrification sludge taken from those SBRs was aerated for 12h, then washed 3 times by deionized water repeatedly, they will be applied to batch tests."
